# The Association between Sedentary Behavior, Physical Activity, and Physical Fitness with Body Mass Index and Sleep Time in Chilean Girls and Boys: A Cross-Sectional Study

**DOI:** 10.3390/children10060981

**Published:** 2023-05-31

**Authors:** Andrés Godoy-Cumillaf, Paola Fuentes-Merino, Claudio Farías-Valenzuela, Daniel Duclos-Bastías, Frano Giakoni-Ramírez, José Bruneau-Chávez, Eugenio Merellano-Navarro

**Affiliations:** 1Grupo de Investigación en Educación Física, Salud y Calidad de Vida (EFISAL), Facultad de Educación, Universidad Autónoma de Chile, Temuco 4780000, Chile; 2Instituto del Deporte, Universidad de las Américas, Santiago 9170022, Chile; 3School of Physical Education, Universidad Católica de Valparaíso, Valparaíso 2340000, Chile; 4IGOID Research Group, Physical Activity and Sport Science Department, University of Castilla-La Mancha, 45071 Toledo, Spain; 5Facultad de Educación y Ciencias Sociales, Instituto del Deporte y Bienestar, Universidad Andres Bello, Santiago 7550000, Chile; 6Departamento de Educación Física, Deportes y Recreación, Universidad de la Frontera, Temuco 4811230, Chile; jose.bruneau@ufrontera.cl; 7Department of Physical Activity Sciences, Faculty of Education Sciences, Universidad Católica del Maule, Talca 3530000, Chile

**Keywords:** cardiorespiratory fitness, strength, speed/agility, flexibility, accelerometer, children, sedentary behavior, sleep, body mass index

## Abstract

Background: In recent decades, the school population has undergone behavioral changes that have affected their health and adult life. The current educational scenario presents high levels of sedentary behavior, physical inactivity, low physical fitness, high levels of obesity, and non-compliance with sleep recommendations. In Chile, the scientific evidence on associations between these behaviors is incipient. Objective: To analyze the association between sedentary behavior, physical activity, and physical fitness with BMI and minutes of sleep in Chilean children aged 10 to 11 years. Methods: A non-probabilistic convenience sample of 222 schoolchildren aged 10 to 11 years. The variables measured were body composition (BMI), cardiorespiratory fitness (20 m shuttle run test), lower and upper muscular strength (long jump test and handgrip dynamometry), speed (4 × 10 m running), and flexibility (sit and reach test). Physical activity and sleep were measured by accelerometers. Results: Of the participants, 60.4% and 90.6% did not comply with sleep and physical activity recommendations, respectively. Physical fitness was higher in boys in all components. The results of the linear regression show that in girls, moderate–vigorous-intensity physical activity, lower- and upper-body muscular strength, and cardiorespiratory endurance were associated with BMI and sleep. In boys, light-intensity physical activity and upper-body muscular strength were associated with both variables. Conclusions: Physical activity intensity, strength, and cardiorespiratory fitness were associated with BMI and sleep; however, physical activity intensity and associated physical fitness components differed by gender.

## 1. Introduction

In the last 40 years, obesity levels have increased considerably, affecting low- and middle-income countries [1]. The causes of this public health problem are diverse, with cultural and behavioral changes in the population caused by globalized development models standing out [2]. Worldwide, regardless of the income levels of the countries studied, obesity has become generalized, while in Latin America there is a worrying and progressive increase [3,4]. In the Chilean context, 58.3% of schoolchildren are overweight or obese [5], and 6% are stunted, which is relevant because it is a chronic indicator of nutritional insufficiency [5].

The behavioral change in the population has led to an increase in sedentary behavior, high rates of physical inactivity, and low levels of physical fitness. In children and young people, sedentary behavior represents 60% of their daily awake time, values that have been increasing in recent years [6]. More time spent carrying out sedentary behavior is associated with health risks, as well as with the development of becoming overweight and obese [7]. Regarding physical inactivity, it is estimated that worldwide, only 20% of children and adolescents comply with the recommended levels of physical activity [8]. In Chile, evidence reports low levels, which tend to decrease with increasing age [9,10]. This situation has become worrisome since evidence indicates that physical inactivity and obesity are risk factors for cardiovascular diseases, some types of cancer [11,12,13], and all causes of premature mortality [14,15], conditions that if acquired in childhood are more likely to be maintained in adulthood [16,17]. To mitigate this, it is recommended that children and young people perform 60 min of moderate–vigorous-intensity physical activity daily [18], since it promotes a reduction in fat mass and body mass index [BMI] [19,20,21]. Regarding physical fitness, worldwide, only 19.3% of children and young people are physically active [8], while in Chile, the evidence indicates that they have low levels [22,23]. This situation is not encouraging, since for the reduction in BMI and total adiposity values, it is necessary to present acceptable to good values [24,25].

On the other hand, with the aim of having healthy habits, 24 h movement pattern recommendations have been elaborated for children and young people, where sedentary behavior and physical activity are complemented with sleep [26]. This position, which highlights the importance of all the behaviors that are performed continuously during the day, results in an increase in one of them causing a decrease in another. Thus, studies in children have shown that physical activity of different intensities throughout the day and sedentary behavior time, mostly in front of screens, are associated with sleep [26,27,28]. As a result, it is recommended that children sleep between 9 and 12 h a day [29].

In Chile, scientific evidence on associations between sedentary behavior, physical activity, and physical fitness with BMI and sleep is incipient [30]. Studies that worked with adolescents [31,32] and university [33,34,35] populations stand out, as they incorporate the measurement of physical activity and sleep objectively. However, it was not possible to find evidence from Chilean studies that collected sleep data in children using accelerometers. This was considered a weakness by the latest Global Matrix 3.0 report [36], as there is a need for objective data to support the design of more feasible and realistic interventions. The effectiveness of objective measurements using accelerometry provides real values of the physical activity and sleep levels of schoolchildren, which increases the quality of research work. Accelerometry is presented as one of the most reliable techniques to record and store the duration, intensity, and frequency of physical activity performed and to give an estimate of the sedentary behavior of children [37], which is why it is necessary to have this type of data in the child and adolescent population since they serve as a basis for the design of more feasible and realistic interventions [38]. On the other hand, it has been shown that from an early age, there are differences between the sexes for sedentary behavior [39], physical activity [39], physical fitness [40], levels of obesity [41], and sleep [42], with women presenting the lowest results in most cases. As a result of all the above arguments, the present research aims to analyze the association between sedentary behavior, physical activity and physical fitness with BMI and minutes of sleep in Chilean boys and girls aged 10 to 11 years.

## 2. Materials and Methods

### 2.1. Study Design

An observational, relational, cross-sectional study was conducted with a quantitative research approach to analyze the association between sedentary behavior, physical activity, and physical fitness with BMI and minutes of sleep in Chilean boys and girls aged 10 to 11 years.

### 2.2. Participants

The sample was non-probabilistic, and for convenience, considering the acceptance to participate in the study and territorial accessibility for the researcher, it was composed of 222 children (130 females and 92 males) aged 10 and 11 years, who were in the fifth grade of primary education in a subsidized private school in Temuco, Chile, which taught two physical education classes per week (90 min each).

The following were considered as inclusion criteria: active enrollment; 10–11 years of age at the time of the study; valid measures of sedentary behavior, physical activity, physical fitness, body mass index, and hours of sleep; written parental consent for their children to participate in the study; and the assent of the participants when they were asked to collaborate. 

Exclusion criteria were those who presented physical and/or mental problems that did not allow them to participate. The anthropometric and physical fitness measurements were performed by graduates in physical education pedagogy, who were trained to guarantee standardization.

This research had the approval of the Scientific Ethical Committee of the Universidad Autónoma de Chile, through the study protocol (N°11-19), as well as the approval of the school. An informed consent form for parents or guardians was used, as well as the assent of the students. 

### 2.3. Procedures and Instruments

The data were collected between the months of April and May 2019.

Body mass index: Weight was measured twice, using light clothing and bare feet, by means of a digital scale (Omrom) with a precision of 0.1 kg. Height was measured twice using a stadiometer (SECA 222), with a precision of 0.1 cm, placing the child barefoot, with the spine aligned with the bar and the chin parallel to the floor. The mean of the weight and height measurements were used to calculate BMI, dividing the weight by the square of the height (kg/m^2^).

Physical fitness: Cardiorespiratory fitness was evaluated using the 20 m shuttle run test, which provides valid and reliable information on maximal aerobic capacity in children [43]. The estimated maximal value of oxygen volume (mL/kg/min) was calculated using the equation proposed by Léger [44]. Lower limb muscle strength was assessed using the feet-together long jump test [43], while upper limb strength was assessed through manual handgrip dynamometry (Takei 5401) [43]. Speed/agility was measured using 4 × 10 m running [43]. Flexibility was measured using the sit and reach test [43].

Sedentary behavior, physical activity, and sleep: These measurements were assessed by means of a GENEActiv triaxial accelerometer, original model (ActivInsights Ltd., Cambridgeshire, UK), which has been shown to provide valid measures of sedentary behavior, physical activity, and sleep [45,46].

The participants were asked to wear the accelerometer for seven consecutive days on their non-dominant wrist. The accelerometers were set to collect data at 100 Hz. The data were downloaded with Geneactiv 2.2 software, using 30 s epochs. Daily minutes carrying out sedentary behavior and light and moderate to vigorous intensity physical activity were determined using cut-off points defined by Phillips et al. [47]. Sleep duration was calculated using the Sadeh algorithm [48], which was implemented in the Excel macro provided by the company Activinsights. Information on sedentary behavior and light-, moderate-, and vigorous-intensity physical activity was provided in average minutes per day. Regarding sleep, the data are provided in minutes of sleep per day.

Regarding compliance with recommendations for the age of the sample evaluated, for sleep, they were between 540 to 720 min (9 to 12 h) [29], while for physical activity, they should accumulate a daily average of 60 min of moderate to vigorous intensity [18].

### 2.4. Statistical Analysis

The data were analyzed using IBM SPSS Statistics version 25 software. The normality of the data was assessed using the Kolmogorov–Smirnov test. 

Gender differences for the variables studied were tested using Student’s *t*-test. The effect size was determined using Cohen’s d. To find out the association between sedentary behavior, physical activity, and physical fitness (in its original measurement scale) with BMI and hours of sleep, multiple linear regression was used, adjusted for gender, age, and weight. Statistical significance was determined at the conventional cut-off point of *p* < 0.05. 

## 3. Results

The characteristics of the participants and the differences by gender are shown in Table 1. For average minutes of sleep, the girls presented lower values than the boys, with significant differences between them and the effect size being small in all cases. All data had a normal distribution.

All the components of physical activity and physical fitness show better results in boys (*p* < 0.05). On average, 90.6% of those evaluated failed to comply with the physical activity recommendations for the child population (60 min of moderate- to vigorous-intensity physical activity per day [18]). More than 60% of those evaluated did not meet the sleep recommendations for their age (9 to 12 h [29]).

Table 2 shows the associations found between sedentary behavior, physical activity, and physical fitness with BMI and minutes of sleep, separated by gender. 

### 3.1. Sedentary Behavior

In girls, sedentary behavior was positively associated with BMI. In boys, sedentary behavior was negatively associated with minutes of sleep.

### 3.2. Physical Activity

In girls, moderate–vigorous-intensity physical activity was negatively associated with BMI, and positively associated with minutes of sleep. In boys, light intensity was positively associated with BMI and negatively associated with minutes of sleep.

### 3.3. Physical Fitness

In girls, upper muscular strength was positively associated with BMI and negatively associated with sleep, while lower-part muscle strength was negatively associated with BMI. Cardiorespiratory fitness was negatively associated with BMI, and positively associated with sleep minutes. In boys, upper-part muscle strength was positively associated with BMI and negatively associated with minutes of sleep. Lower part muscle strength and speed/agility were positively associated with minutes of sleep. Cardiorespiratory fitness was negatively associated with BMI.

## 4. Discussion

This research aimed to analyze the association between sedentary behavior, physical activity, and physical fitness with BMI and minutes of sleep in Chilean boys and girls aged 10 to 11 years. The results of this study show that linear regression showed associations between the intensity of physical activity with some components of physical fitness and sleep; however, this was different according to gender. In addition, a large part of the population evaluated did not meet the recommendations for physical activity and minutes of sleep for their age.

In this research, a low percentage of those evaluated complied with the daily recommendations for moderate–vigorous physical activity (9.4%), results that are lower than those reported in studies in foreign populations [49,50,51]. These differences could be explained, among other factors, by the long days of low energy expenditure that schoolchildren have at school [52], which in the case of Chile, can exceed nine hours of school hours, and the misuse of electronic devices, which increase sedentary behavior [53]. On the other hand, there are studies that indicate that Chilean parents have low levels of physical activity and physical fitness [54], with the parental relationship being considered relevant in the transmission of behavior and models to be followed by their children [55] and therefore, the physical activity that children perform is related to that of their parents [56,57]. The results also present statistically significant differences between boys and girls in sedentary behavior and light and moderate–vigorous physical activity, with girls having lower levels compared to boys. This is consistent with a recent study conducted in public schools in the country [58] and with international studies [59,60], where the negative influence of gender stereotypes on the participation and practice of physical activity is noted.

In relation to minutes of sleep, statistically significant differences were found between girls and boys (*p* = 0.046), with girls having less time per day, results that agree with similar studies [61]. In relation to sleep time compliance, 39.6% of those evaluated complied with the recommendations, values that are lower than those previously reported in a population of similar age [62]. A recent study using questionnaires to determine sleep problems in a school population found that 81.9% of schoolchildren had sleep problems, increasing the likelihood of memory problems, slowness in mathematical problems, difficulty in maintaining attention, solving complex problems, and nervousness [63]. However, it should be noted that all studies in the Chilean context describe this variable through subjective instruments, so the present results through accelerometry have greater objectivity.

In this context, at the international level, some studies have reported a lack of sleep time in children [64], which is a worrying situation, since it makes children more susceptible to feeling tired and sleepy the next day, alterations in their mood [65], and academic performance problems [66]. It has been evidenced that chronic lack of sleep since childhood increases the risk of developing cardiovascular and metabolic diseases [67,68,69] with a doubled risk of being overweight or obese [70].

The results of the linear regression show that in girls, sedentary behavior has a high β of 0.491, positively influencing BMI (CI = 0.001–0.652), an expected result, because the greater the sedentary behavior, the lower the energy expenditure, causing an increase in adiposity [71]. Regarding the intensity of physical activity, moderate–vigorous-intensity physical activity in girls was negatively associated with BMI, while light-intensity physical activity was positively associated with BMI in boys. This result may be explained by the large difference in total physical activity and intensity in boys, almost doubling that of girls. The association between physical activity intensity and BMI is in line with cross-sectional studies, where they have observed that this intensity of physical activity reduces some measures of adiposity such as BMI or total fat mass [19,20,21,72]; however, these results should be analyzed with caution due to an anthropometric factor, since the weight of the child influences the intensity and frequency of physical activity, a situation that cross-sectional studies cannot analyze or because the studies differ in the instruments used to evaluate physical activity (questionnaires and accelerometry) or adiposity indices (BMI, body perimeters, skinfolds, and DXA) [73].

With respect to the components of physical fitness in girls and boys, upper muscular strength had a positive association with BMI, and cardiorespiratory fitness was negatively associated with BMI, situations that have been previously reported [19,74,75,76] and which support strength and cardiorespiratory fitness as indicators of health in children. However, in girls, the results also show a negative association of lower-part muscle strength with BMI, which could be due to biological factors, specifically body composition, since at the age of the population evaluated, girls have more adipose tissue in their bodies [77], less physical activity, and lower levels of physical fitness [78,79,80]. On the other hand, in girls, moderate–vigorous-intensity physical activity was positively associated with minutes of sleep, which is similar to what has been previously reported [81] but different from what was reported in a study that worked with Spanish children, which found that the same type of intensity of physical activity was associated with a decrease in hours of sleep [82]. In this sense, a systematic review that analyzed how exercise impacts sleep duration reported that the results vary significantly due to exercise intensity, but it mostly reported an improvement, concluding that exercise promotes longer sleep duration [83]. Regarding physical activity intensity, some studies report that light intensity brings better results for sleep [81,82], while other studies report that moderate–vigorous-intensity physical activity has greater benefits [84,85]. In relation to this, a meta-analysis on the effects of physical activity on sleep found no significant differences between intensities [86], given that the effects of exercise intensity on sleep time are varied, so further research is needed to explore the biological mechanisms that modulate the dynamic interaction between these two aspects [83]. Finally, the results of the associations between the components of physical fitness and minutes of sleep are different by gender, although in both there was an association in upper-body muscle strength; in girls, an association was found with cardiorespiratory fitness, and in boys, an association was found with lower-body muscle strength and speed. This may be due to the fact that good physical fitness values are factors that could benefit sleep, a situation that could be explained because a greater amount of sleep is associated with an increase in IGF-1, which is an anabolic hormone that plays an important role in the maintenance of muscle mass [87], and that on the contrary, a lack of sleep decreases the activity of the protein synthesis pathways and increases the activity of the degradation pathways, favoring the loss of muscle mass and thus hindering muscle recovery [88]. It is important to note that not many studies have studied the strength–sleep relationship in children, so further epidemiological research and intervention studies are needed to determine whether these relationships manifest.

The findings of this study highlight the importance of physical activity and physical condition in health parameters, which can be used as a means for the prevention and treatment of the state of being overweight, obesity, and sleep problems in children.

Although the children evaluated do not correspond to a representative sample, which is why the results cannot be generalized, the findings of this research may constitute a basis for decision-making aimed at improving the health of the child population. It is here where the implementation of an action plan between parents and schools becomes necessary since the former directly influences the life habits of the children and the latter also plays an important role since it has been shown that strategies that have been implemented to increase the practice of physical activity have been effective.

One limitation of this study is its cross-sectional design, which prevents a better understanding of the phenomenon and the drawing of cause–effect inferences; nevertheless, it presents objective data in the Chilean school population, which is a need stated by different studies. Another limitation is not having measured variables such as food and the use of electronic media, which directly influence BMI and sleep time. The non-probabilistic sample, which comes from a single region of the country, does not allow inferences to be made about all Chilean children. In addition, the relative effect of age [89] or biological maturity, factors that influence physical fitness, was not determined in the sample, which should be considered in future research. The strengths of this study are the standardization of the physical fitness measurements, the objectivity of the physical activity and sleep measurements, and the inclusion of a population that has been scarcely studied, which may be useful for future related research.

## 5. Conclusions

The results of this study show that, in girls, moderate–vigorous-intensity physical activity, lower- and upper-body strength, and cardiorespiratory endurance were associated with BMI and sleep. In boys, light-intensity physical activity and upper-body muscular strength were associated with both variables. In addition, a low percentage of the population evaluated did not meet the recommendations for physical activity and sleep time for their age.

This study highlights the need for a concerted action plan between parents and teachers to increase the physical activity undertaken by children, thus favoring the overall health of the student. To better understand the findings, longitudinal and intervention studies that analyze the relationship between moderate–vigorous-intensity physical activity and strength with minutes of sleep are needed. In addition, the findings suggest that the implementation of interventions to improve physical fitness in children could be an effective strategy to reduce levels of sedentary behavior, states of being overweight, and obesity.

## Figures and Tables

**Table 1 children-10-00981-t001:** Characteristics of the sample by total and gender.

		Total (*n* = 222)	Girls (*n* = 130)	Boys (*n* = 92)	*p*	ES
Anthropometric characteristics						
Weight (kg)		45 (9.7)	45.4 (10.3)	44.5 (9)	0.504	-
Height (cm)		147 (7.2)	146.6 (8.5)	147.7 (4.9)	0.299	-
BMI (kg/m^2^)		20.6 (2.9)	20.8 (2.8)	20.2 (3)	0.121	-
Sleep						
Sleep (min)		552.4 (125.2)	538.3 (132.3)	572.4 (168.1)	**0.046**	0.18
Compliance with recommendations (%)	Comply	39.6	38.4	41.1	**-**	-
	Did not comply	60.4	61.6	58.7	**-**	-
Sedentary behavior		800.7 (110.1)	815.6 (70.8)	779.9 (146.9)	**0.016**	0.21
Physical activity						
Light (min)		21.1 (8.2)	18.9 (7)	24.2 (8.8)	**0.000**	0.16
Moderate–vigorous (min)		31.1 (19.9)	23 (13.5)	43.1 (21.5)	**0.000**	0.22
Compliance with recommendations (%)	Comply	9.4	1.5	20.6	**-**	-
	Did not comply	90.6	98.5	79.4	**-**	-
Physical fitness						
Muscular strength upper part (kg)		16.6 (3.3)	15.8 (2.6)	17.7 3.8)	**0.000**	0.22
Muscle strength lower part (cm)		119.5 (17.9)	112 (13.7)	129.8 (18.1)	**0.000**	0.16
Speed/agility (s) ^¥^		14.3 (1.4)	14.8 (1.2)	13.6 (1.3)	**0.000**	0.32
CRF (mL/kg/min)		44.3 (3.8)	42.8 (2.8)	46.5 (4)	**0.000**	0.21

CRF, cardiorespiratory fitness. ES, effect size (Cohen’s d). ^¥^ Less time (in s) indicates better speed/agility levels. The values in bold indicate a statistical significance for *p* < 0.05.

**Table 2 children-10-00981-t002:** Association between physical activity and physical fitness, with BMI and minutes of sleep.

	BMI			SLEEP		
	*B*	CI 95%	Sig	*B*	CI 95%	Sig
Girls (*n* = 130)						
Sedentary behavior	0.491	0.001–0.652	**0.000**	0.09	−0.013–0.346	0.297
Physical activity						
Light	0.139	−0.022–0.192	0.148	−0.207	−4.080–1.697	0.072
Moderate–vigorous	−0.278	−0.365–−0.017	**0.017**	0.399	−0.520–2.938	**0.005**
Physical fitness						
Muscular strength upper part	0.247	0.210–0.487	**0.008**	−0.26	−2.443–−0.050	**0.020**
Muscular strength lower part	−0.193	−0.296–−0.042	**0.042**	0.125	0.028–3.289	0.272
Speed/agility ^¥^	−0.255	−0.653–0.124	0.069	0.055	−3.328–39.579	0.695
CRF (mL/kg/min)	−0.539	−0.700–−0.067	**0.000**	0.249	−4.673–8.193	**0.006**
Boys (*n* = 92)						
Sedentary behavior	−0.07	−0.080–−0.002	0.417	−0.238	−0.346–0.013	**0.017**
Physical activity						
Light	0.392	−0.022–0.592	**0.000**	−0.425	−4.080–1.697	**0.001**
Moderate–vigorous	−0.206	−0.365–−0.017	0.067	0.093	−0.520–2.938	0.465
Physical fitness						
Muscular strength upper part	0.428	0.670–0.387	**0.000**	−0.656	−2.443–−0.450	**0.000**
Muscular strength lower part	−0.200	−0.296–−0.042	0.149	0.493	0.328–3.289	**0.002**
Speed/agility ^¥^	−0.333	−0.653–0.124	0.086	0.920	0.328–3.579	**0.000**
CRF (mL/kg/min)	−0.325	−0.400–−0.067	**0.032**	0.253	−4.673–8.193	0.141

CRF, cardiorespiratory fitness. ^¥^ Less time (in s) indicates better speed/agility levels. The values in bold indicate a statistical significance for *p* < 0.05.

## Data Availability

The data presented in this study are available from the corresponding author upon request.

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
