# Peer review of "The Association between Sedentary Behavior, Physical Activity, and Physical Fitness with Body Mass Index and Sleep Time in Chilean Girls and Boys: A Cross-Sectional Study"

_children, 2023, doi:10.3390/children10060981_

Round 1

Reviewer 1 Report

SUMMARY

The study investigated the association between sedentary behaviour, physical activity and physical fitness with BMI and minutes of sleep in Chilean children aged 10 to 11 years. Authors found that associations between physical activity intensity with some components of physical fitness and sleep, however, this was different according to gender.

I congratulate authors on their work. The topic is timely and important.  However, the article lacks on certain key aspects therefore before this can be considered for publication, I suggest authors to consider my points below.

TITLE

Please add what was the study design.

ABSTRACT

The link between background information and study objective is not clear.

Please add details how did you measure other outcomes.

Line 29: “men”; in the rest of the paper you use “boys”

Conclusions are just a repetition of the results. Please state the implications of the findings. 

INTRODUCTION 

Please emphasise in your introduction: (i) what are the shortcomings of previous research that necessitates the need for your study, (ii) what will be the added value of your study to the body of knowledge, and (iii) who will benefit.

Line 76: using accelerometers is central point of the study. As such there should be more discussion around accelerometers here. For example, authors could introduce the readers unfamiliar with accelerometers to what they are capable of for example: joint angle estimations https://www.ncbi.nlm.nih.gov/pmc/articles/PMC3175383/ , activity classification https://www.nature.com/articles/s41598-022-18845-x , fall prediction https://www.sciencedirect.com/science/article/pii/S2665917422002483

METHODS

The whole methods section should be rewritten (where needed: see detailed comments below) according to relevant reporting guidelines for observational studies, such as STROBE. Information on research question, study design, setting, inclusion and exclusion criteria for study participants, the definition of outcomes, validity and reliability of instruments, data postprocessing, sample size estimation, data collection procedures, ethics etc is either missing or very superficially described and should be provided in subsections to facilitate reading.

Please start the methods section with specifying the research question and follow by specifying what was the study design. 

Sample size calculation is not provided. Lack of justification why sample size calculation was not performed. 

More details are needed regarding recruitment procedures. From where did you recruit participants, in what time frame and what was the recruitment rate that how many invited and how many rejected the invite or were not eligible etc. 

How did you ensure ethical aspects were considered? There is no mention of the need of signing consent forms, data protection aspects, or whether you used any participant information sheets. 

Please provide data on technical specification of the instruments used and their validity/reliability in regard to measuring all the outcomes, where it is absent.

How did you address statistical bias due to the issue of multiple testing 

RESULTS

Please start off by stating the results of the data distribution normality checks. 

To improve readability, please avoid repeating the same information in the main text that is already in the tables (i.e. numbers). 

Given the amount of information, to improve readability, please add subsection based on the study aims or study outcomes.

DISCUSSION 

I suggest starting off the discussion with reminding the reader about the study objectives.

Please discuss the potential impact of your findings on research in the field.

Please discuss how generalizable findings are and whether there are any potential clinical implications.

Author Response

SUMMARY

The study investigated the association between sedentary behaviour, physical activity and physical fitness with BMI and minutes of sleep in Chilean children aged 10 to 11 years. Authors found that associations between physical activity intensity with some components of physical fitness and sleep, however, this was different according to gender.

I congratulate authors on their work. The topic is timely and important.  However, the article lacks on certain key aspects therefore before this can be considered for publication, I suggest authors to consider my points below.

TITLE

Please add what was the study design.

A: Aggregate.

ABSTRACT

The link between background information and study objective is not clear.

A: Information was added that allows linking background information with the objective of the study (lines 23 and 24).

Please add details how did you measure other outcomes.

A: Aggregate (lines 27 to 29).

Line 29: “men”; in the rest of the paper you use “boys”

A: Modified.

Conclusions are just a repetition of the results. Please state the implications of the findings. 

A: Modified (lines 34 to 37).

INTRODUCTION 

Please emphasise in your introduction: (i) what are the shortcomings of previous research that necessitates the need for your study, (ii) what will be the added value of your study to the body of knowledge, and (iii) who will benefit.

A: We have added information as recommended (lines 79 to 85).

Line 76: using accelerometers is central point of the study. As such there should be more discussion around accelerometers here. For example, authors could introduce the readers unfamiliar with accelerometers to what they are capable of for example: joint angle estimations https://www.ncbi.nlm.nih.gov/pmc/articles/PMC3175383/ , activity classification https://www.nature.com/articles/s41598-022-18845-x , fall prediction https://www.sciencedirect.com/science/article/pii/S2665917422002483

A: We have added information as recommended (lines 85 to 87).

METHODS

The whole methods section should be rewritten (where needed: see detailed comments below) according to relevant reporting guidelines for observational studies, such as STROBE. Information on research question, study design, setting, inclusion and exclusion criteria for study participants, the definition of outcomes, validity and reliability of instruments, data postprocessing, sample size estimation, data collection procedures, ethics etc is either missing or very superficially described and should be provided in subsections to facilitate reading.

Please start the methods section with specifying the research question and follow by specifying what was the study design. 

Sample size calculation is not provided. Lack of justification why sample size calculation was not performed. 

More details are needed regarding recruitment procedures. From where did you recruit participants, in what time frame and what was the recruitment rate that how many invited and how many rejected the invite or were not eligible etc. 

How did you ensure ethical aspects were considered? There is no mention of the need of signing consent forms, data protection aspects, or whether you used any participant information sheets. 

Please provide data on technical specification of the instruments used and their validity/reliability in regard to measuring all the outcomes, where it is absent.

How did you address statistical bias due to the issue of multiple testing 

A: Thanks for his comment. We have restructured the methods section as recommended.

RESULTS

Please start off by stating the results of the data distribution normality checks. 

A: Aggregate (line 168)

To improve readability, please avoid repeating the same information in the main text that is already in the tables (i.e. numbers). 

A: According to the suggestions, in the text we have eliminated the numbers.

Given the amount of information, to improve readability, please add subsection based on the study aims or study outcomes.

A: Thank you very much for your comment, it allowed us to give a better understanding of our results. We have divided the wording of results according to the variables studied (lines 182 to 196).

DISCUSSION 

I suggest starting off the discussion with reminding the reader about the study objectives.

A: Aggregate (lines 200 to 202).

Please discuss the potential impact of your findings on research in the field.

A: According to the recommended information was added (lines 287 to 289).

Please discuss how generalizable findings are and whether there are any potential clinical implications.

A: According to the recommended information was added (lines 290 to 296).

Reviewer 2 Report

 General comments

The purpose of the present study was to analyze the association between sedentary behavior, physical activity and physical fitness with BMI and minutes of sleep in Chilean children aged 10 to 24 years. The authors found that 60.4% and 90.6% did not comply with sleep and physical activity recommendations and. physical fitness was higher in boys compared to girls in all components. The authors also conducted a linear regression which showed that physical activity intensity is associated with some components of physical fitness and sleep and this relationship was different according to gender. In my opinion such studies are very important for the development of children – students because we are living in the era of technology which decreases the physical activity and thus body composition and physical fitness. The authors did a nice job, however there are some issues that should be addressed and I also have some recommendations to the authors:

1.      The authors used a sample of 10 and 11 year old children which is a large time frame. For example if most boys was 11 year old and especially was born in the last semester of the year (relative age effect) then it is possible to find differences in physical fitness. Also did you measure biological maturity (peak height velocity?). Please provide specific ages of both boys and girls. Furthermore it will add more to the paper if you could provide the means and standard deviations of peak height velocity.

2.      I also recommend to calculate the percentiles for the BMI by using the LMS method as previously described (Flegal and Cole, 2013; Flegal KM, Cole TJ. Construction of LMS parameters for the Centers for Disease Control and Prevention 2000 growth charts. Natl Health Stat Report, 2013; 63: 1–3). It will be useful especially in the discussion section and will provide the level of BMI of Chilean children.

3.      Another recommendation is to provide the effect size Cohen’s d for pairwise comparisons between sexes (Table 1).

Specific comments

Abstract

Please provide more important results in the abstract.

Introduction

I recommend to the authors to mention some papers in a small paragraph the differences between sexes for 10-11 year old children about sedentary behavior, physical activity, physical fitness, BMI and sleep in the intro.

Statistics-results

I recommend to the authors to calculate the effect size Cohen d for pairwise comparisons between sexes (for Table 1).

Please add the type of multiple regression that you used (e.g standard, hierarchical, stepwise?).

Page 4 lines 151-152: You mention that: “All the components of physical activity and physical fitness show better results in children (p<0.05)” Do you mean “better results in boys?”.

Author Response

The purpose of the present study was to analyze the association between sedentary behavior, physical activity and physical fitness with BMI and minutes of sleep in Chilean children aged 10 to 24 years. The authors found that 60.4% and 90.6% did not comply with sleep and physical activity recommendations and. physical fitness was higher in boys compared to girls in all components. The authors also conducted a linear regression which showed that physical activity intensity is associated with some components of physical fitness and sleep and this relationship was different according to gender. In my opinion such studies are very important for the development of children – students because we are living in the era of technology which decreases the physical activity and thus body composition and physical fitness. The authors did a nice job, however there are some issues that should be addressed and I also have some recommendations to the authors:

  1. The authors used a sample of 10 and 11 year old children which is a large time frame. For example if most boys was 11 year old and especially was born in the last semester of the year (relative age effect) then it is possible to find differences in physical fitness. Also did you measure biological maturity (peak height velocity?). Please provide specific ages of both boys and girls. Furthermore it will add more to the paper if you could provide the means and standard deviations of peak height velocity.

A: We appreciate your comment. The relative effect of age and biological maturity directly influence physical fitness, unfortunately when reviewing our database we do not have the information on dates of birth, we only collect the age they were at the time of being evaluated. Due to the importance of his comment, we add information on the limitations of the research (lines 284 to 286), in addition to taking it into consideration for future research.

  1. I also recommend to calculate the percentiles for the BMI by using the LMS method as previously described (Flegal and Cole, 2013; Flegal KM, Cole TJ. Construction of LMS parameters for the Centers for Disease Control and Prevention 2000 growth charts. Natl Health Stat Report, 2013; 63: 1–3). It will be useful especially in the discussion section and will provide the level of BMI of Chilean children.

A: We appreciate your comment. Together with the other authors of the manuscript, we discussed the recommendation to calculate BMI percentiles using the LMS method. Since the objective only has the association of variables, we have used the BMI in its original value, which is why on this occasion we have not contemplated its recommendation, but it will be considered for future publications that will address this topic.

  1. Another recommendation is to provide the effect size Cohen’s d for pairwise comparisons between sexes (Table 1).

A: Aggregate.

Specific comments

Abstract

Please provide more important results in the abstract.

A: Aggregate (lines 31 to 34)

Introduction

I recommend to the authors to mention some papers in a small paragraph the differences between sexes for 10-11 year old children about sedentary behavior, physical activity, physical fitness, BMI and sleep in the intro.

A: Aggregate (Lines 90 and 91)

Statistics-results

I recommend to the authors to calculate the effect size Cohen d for pairwise comparisons between sexes (for Table 1).

A: Aggregate

Please add the type of multiple regression that you used (e.g standard, hierarchical, stepwise?).

A: Aggregate.

Page 4 lines 151-152: You mention that: “All the components of physical activity and physical fitness show better results in children (p<0.05)” Do you mean “better results in boys?”.

A: Thank you for your comment. Modified.

Round 2

Reviewer 1 Report

Authors full addressed my comments.

Reviewer 2 Report

The authors have significantly improved the quality of the manuscript and satisfy most of my concerns. Congratulations to the authors!!